# Analysis of Bubble Flow Mechanism and Characteristics in Gas–Liquid Cyclone Separator

**Yujie Bai [1,2], Hong Ji [1,2,\*], Yaozhuo Liu [1,2], Lei Li [1,2] and Shengqing Yang [1,2]**

1 Energy and Power Engineering School, Lanzhou University of Technology, Lanzhou 730050, China; 13754880446@163.com (Y.B.); lyzh18119343073@163.com (Y.L.); lut2489883853@163.com (L.L.); yangsq@liugong.com (S.Y.)
2 Gansu Hydraulic and Pneumatic Engineering Technology Research Center, Lanzhou 730050, China
\* Correspondence: jihong@lut.cn; Tel.: +86-138-9322-5310

**Abstract:** The separation of bubbles in a gas–liquid cyclone is complicated. A combination of numerical simulation and visual experimentation was considered apt to reveal the microscopic mechanisms of bubble flow. First of all, cyclones with different structures were numerically simulated. The calculation results show that the larger the diameter of the exhaust port, the better the bubble flow effect. When the exhaust port diameter was 24 mm, the gas discharge efficiency was 8% higher than that with an exhaust port diameter of 16 mm. The sequence of the bubble flow effect of a four-structure cyclone was obtained, and the gas discharge efficiency of the cyclone with a rectangular inlet was 7% higher than that of the trapezoidal inlet. Finally, a visual experimental platform was built to compare the rectangular inlet cyclone and spiral inlet cyclone with the best bubble flow effect. In accordance with the simulation numerical calculations, the bubble flow effect of the rectangular inlet cyclone was better than that of the spiral and trapezoid inlet cyclones, and the rectangular inlet in the middle was better. This article provides a specific theory and experience to guide further research on the separation mechanism, flow field characteristics and structurally optimal design of gas–liquid cyclones.

**Keywords:** gas–liquid cyclone; bubble flow; numerical simulation; experimental research

## 1. Introduction

The gas–liquid cyclone separator is a gas–liquid separation device commonly used in petrochemical, food and other fields. It has the advantages of a simple structure, high degassing efficiency, small installation space and simple and convenient operation. Due to the numerous hazards caused by the gas contained in the hydraulic oil, the problem of oil degassing has been paid more and more attention by scholars at home and abroad in recent years. The degassing effect of the gas–liquid cyclone separator in the hydraulic system is obvious, and it has broad prospects for engineering applications [1–3]. Although the structure of the cyclone is simple, the internal flow field is more complicated. Compared with the gas–solid and liquid–solid two-phase flow, the internal gas–liquid two-phase flow is more complicated due to the instability of the liquid [4,5], and there are also more uncertain calculated factors, which means that theoretical research into gas–liquid cyclone separation lags behind that on cyclones and hydrocyclones [6,7].

The cyclone is a device that uses the principle of centrifugal sedimentation to separate multiple phases in a flow field. When the cyclone is working, the multiphase mixed liquid enters the cyclone from the feed port tangentially at a certain speed, forming a fast double layer swirling flow, in which the outer layer is swirling downward and the inner layer is swirling upward [8]. After the gas–liquid mixture flows tangentially from the cyclone, the mixture rotates in the cyclone to separate the centrifugal force from gravity. The denser liquid flows out along the inner wall of the cyclone to the bottom swirl port, and the less dense gas is separated in the center of the swirl to form an inverted cone-shaped

swirl surface, finally leaving via the top overflow port of the swirler exhaust; gas–liquid two-phase cyclone separation is thereby achieved [9].

Numerical and experimental studies on the structural parameters of the cyclone and its degassing efficiency have been conducted [10–13]. The research shows that the bubbles mixed into the oil can be effectively eliminated with an appropriate design for the structure of the cyclone separator. Because the bubble separator uses oil rotation to make the bubbles gather towards the center [14–17], no external power is needed. The equipment, the bubble separator, has an obvious de-bubbling effect on the gas-containing oil.

Li Qiang used numerical simulation and experimental methods to evaluate the performance of the notched cyclone in the tangential inlet cyclone [18]. Qing Wei used the Reynolds stress model to simulate the flow field of cyclones with different cyclone diameters and inlet sizes to determine the mechanism of axial velocity stagnation [19–21]. Wei Pengkai designed a new type of high-speed wet gas–liquid pipeline separator and concluded that the separation efficiency increases with the gas–liquid superficial velocity [22]. Zhou Wen used a pressure sensor and a five-hole probe to measure the pressure and velocity distribution of the gas flow and proposed an improved weighting method for calculating the droplet separation efficiency [23]. Zeng Xiaobo designed a new type of gas–liquid separator that combines gravity separation and centrifugal separation, and concluded that as long as the liquid level in the liquid pipe is at a reasonable position, the separator can maintain efficient separation under multiple flow patterns [24]. Luo Xiaoming implemented a slug flow simulation method based on the volume of fluid (VOF) multiphase flow model through a custom function (UDF) in ANSYS, and applied it to the diffusion of liquid slugs in the inlet pipe of a gas–liquid cylindrical cyclone (GLCC). When changing the expansion ratio and the inclination angle at the same time, it was concluded that the increase in the inclination angle will reduce the gas–liquid carrier gas depth but enhance the gas–liquid mixing and increase the liquid carrier gas depth [25]. Wang Gang conducted experimental research on the swirl vane gas–liquid separator, and studied the influence of the gas content, Reynolds number and flow adjustment elements on the separation performance, concluding that the increase in Reynolds number and the arrangement of flow-regulating elements can only suppress the oscillation of the air core to a certain extent, and that the separator is more suitable for operation under low air content [26]. Li Yiqian designed a new type of gas–liquid separator. The purpose of using this equipment is to alleviate the problem of the liquid entrainment of natural gas in the compressor, increase the service life of the compressor, and reduce operation and maintenance costs [27–30]. A consultation of the related literature revealed little research on gas–liquid cyclones with different structures and the visualization of bubble flow conditions. This research provides theoretical support for the further optimization of the structure of gas–liquid cyclones and used the visualization of bubble flow conditions. The experiment verifies the feasibility of the relevant structural design and the rationality of the relevant simulation results.

This study uses a combination of numerical simulation and visual experimentation to study the bubble flow states in cyclones with different structures. Through observations with high-speed cameras and the results of numerical calculations, the flow law for bubbles in a cyclone separator is revealed.

## 2. Simulation Analysis

### 2.1. Cyclone Separator Model

Models of cyclones with different structures were created with reference to experimental models. Figure 1 shows the schematic diagrams of cyclones with different structures. The specific parameters are shown in Table 1. Then their flow channels were meshed. In order to obtain high-precision calculations and good convergence, a structured grid was used for division, and the grid independence was tested; the output grid was then input into Fluent, and its boundary conditions were set. The response surface method (RSM) turbulence model was used to complete the numerical calculation of the cyclone. Through the numerical calculation of the pressure field, velocity field, turbulence field, etc., the

internal velocity, pressure and turbulence field distribution rules for different structures of cyclones were obtained, so as to better elucidate the flow of bubbles in the cyclones.

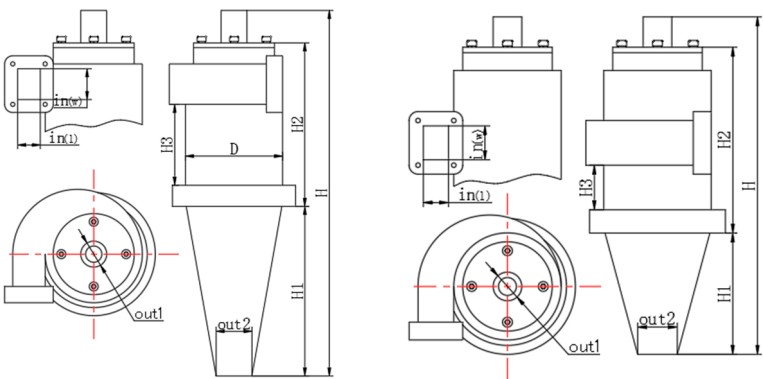

(**a**) Rectangular inlet cyclone  (**b**) Rectangular central inlet cyclone

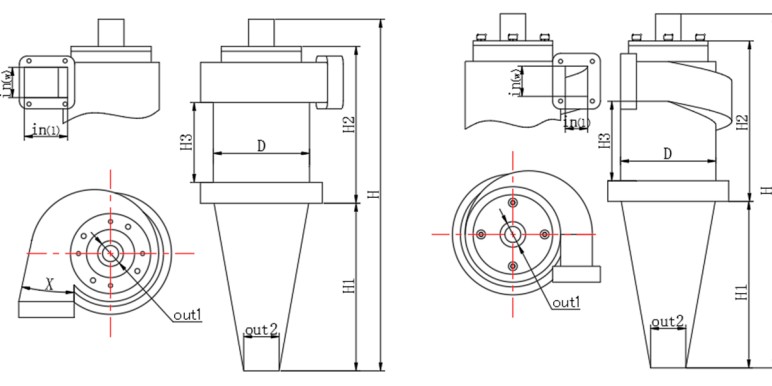

(**c**) Trapezoidal inlet cyclone  (**d**) Spiral inlet cyclone

**Figure 1.** Structure diagram of cyclone.

**Table 1.** Geometry of the cyclone.

| Dimension | Scale (mm) | | | |
|---|---|---|---|---|
| | **Trapezoidal** | **Spiral** | **Rectangular** | **Rectangular Central** |
| in′ | 19 | 19 | 19 | 19 |
| in″ | 27 | 14 | 14 | 14 |
| X | 12° | 0° | 0° | 0° |
| D | 60 | 60 | 60 | 60 |
| H | 220 | 223 | 226 | 189 |
| H1 | 105 | 105 | 105 | 68 |
| H2 | 98 | 101 | 101 | 164 |
| H3 | 50 | 50 | 50 | 25 |
| out1 | 16 | 16 | 16 | 16 |
| out2 | 22 | 22 | 22 | 22 |

Figure 2 is the irrelevance test diagram under five types of grid cells. It can be seen from the figure that the degassing rate was the lowest when the grid cells number was 1,350,000. The degassing efficiency also increased with an increase in the level. The grid cells number of 1,450,000 resulted in the best rate, but too many grid cells not only increased the calculation time but even reduced the separation efficiency. The separation efficiency was the best when the number of grid cells was about 1,450,000. The number of grid cells in this study was 1,449,887.

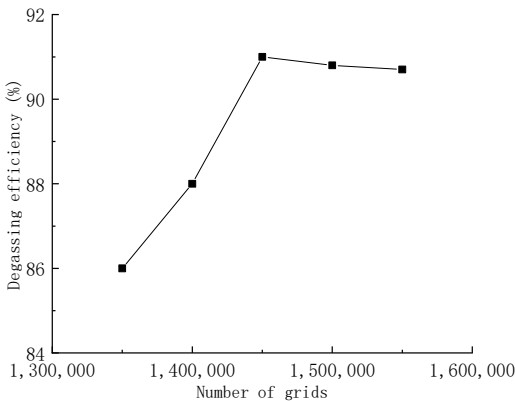

**Figure 2.** Grid independence test graph.

The three-dimensional flow field of the adiabatic cyclone was simulated by computational fluid dynamics (CFD). For a stable incompressible flow, the continuity, momentum and RSM model equations are as follows:

$$\frac{\partial \overline{u_j}}{\partial \overline{x_j}} = 0 \tag{1}$$

$$\overline{u_i}\frac{\partial \overline{u_j}}{\partial x_j} = -\frac{1}{\rho}\frac{\partial \overline{P}}{\partial x_j} + \frac{1}{\rho}\frac{\partial}{\partial x_j}\left[ \mu\left( \frac{\partial U_i}{\partial x_j} + \frac{\partial U_j}{\partial x_i} \right) - \rho\overline{u_i'u_j'} \right] \tag{2}$$

$$\frac{\partial\left(\rho\overline{u_i'u_j'}\right)}{\partial t} + \frac{\partial\left(\rho u_k\overline{u_i'u_j'}\right)}{\partial x_k} = \frac{D\left(\rho\overline{u_i'u_j'}\right)}{Dt} = D_{i,j} + P_{i,j} + G_{i,j} + \Phi_{i,j} - \varepsilon_{i,j} + F_{i,j} \tag{3}$$

In the above formula, $x_j$ represents the coordinate position, $u_j$ is the time-average velocity component, $\overline{P}$ is the time-average pressure, $\mu$ is the molecular viscosity, $\rho$ is the fluid density, and $\overline{u_i'u_j'}$ is the unknown Reynold stress component determined by the turbulence model.

### 2.2. Boundary Conditions and Numerical Schemes

There are three multiphase flow models in Fluent: the VOF, Mixture and Euler models. Compared with the Euler model, Mixture is more suitable for an environment with a strong swirling multiphase flow. The Mixture model was selected in this study for comprehensive comparison. The (SIMPLE) algorithm was used for the pressure-velocity coupling form, (PRESTO) was used for the pressure compensation format, and (QUICK) was used for the discrete momentum.

The RSM model was chosen for the turbulence model because it has a better effect on the multiphase flow and strong swirl environment. The setting of the boundary conditions was as follows: (1) The inlet speed of the two phases was the same, set to 8 m/s. (2) The inlet was set as the speed inlet, and the exhaust port and the liquid discharge port were both set as the pressure outlet. (3) The wall was treated as a non-slip boundary condition, that is, the velocity and turbulence intensity were both zero.

### 2.3. Analysis of Simulation Results

In order to study the flow of bubbles in cyclones with different structures, and to study the distribution of internal velocity, pressure and turbulence fields, the following was carried out: (1) Keeping the other conditions unchanged, three types of rectangular central inlet cyclones with port diameters of 16, 20 and 24 mm were used for simulation analysis; (2) the rectangular inlet cyclone and trapezoidal inlet cyclone, whose inlet shapes are rectangular and trapezoidal, and the spiral inlet cyclone, whose inlet mode is spiral,

were chosen for simulation analysis; (3) the rectangular cyclone and rectangular central cyclone, with the other conditions unchanged, were selected for simulation analysis.

The discharge port was selected as the height reference (Z = 0), the direction of the exhaust port was the positive height direction, and a section Z that affects the separation structure was selected as the analysis section, where section Z = 60 mm.

In order to explore the influence of the diameter of the exhaust port on the bubble movement, the first scheme explored three cases where the diameter of the exhaust port ($D_{(out1)}$) was designed to be 16, 20 or 24 mm to explore its impact on the internal flow field.

It can be concluded from Figure 3 that the pressure in the center area of the exhaust port and the discharge port of the cyclone was the lowest, and the pressure at the inlet was the highest; the internal pressure was distributed symmetrically along the central axis, gradually decreasing from the outside to inside. The larger the diameter of the exhaust port, the smaller the pressure drop, which means that the smaller the size of the exhaust port, the more the internal liquid accumulation caused the pressure drop to increase. It can be seen from Figure 4 that the pressure difference was the largest when $D_{(out1)}$ =16 mm, and the overall pressure was the smallest among the three structures when $D_{(out1)}$ = 24 mm; as the diameter of the exhaust port increased, the pressure gradient gradually decreased, and due to the increase in $D_{(out1)}$, the static pressure and pressure loss gradually decreased.

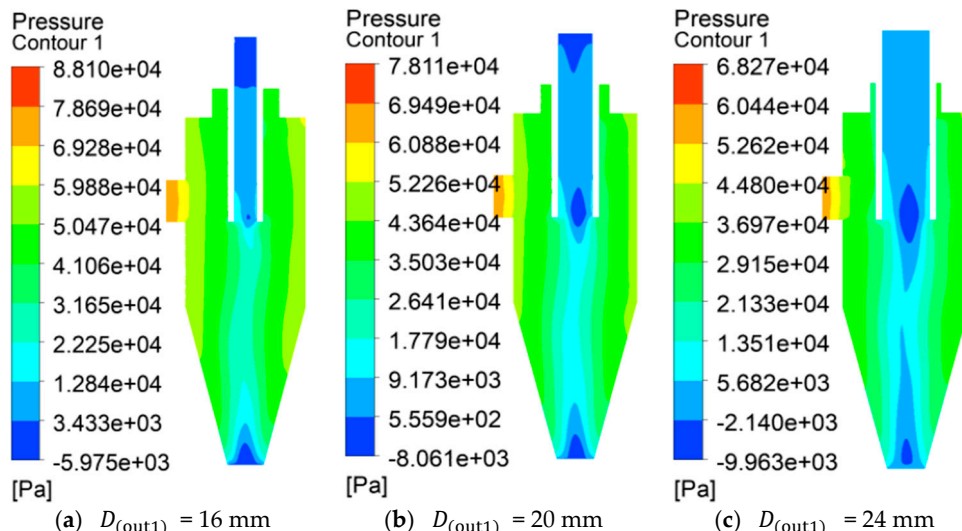

(**a**) $D_{(out1)}$ = 16 mm     (**b**) $D_{(out1)}$ = 20 mm     (**c**) $D_{(out1)}$ = 24 mm

**Figure 3.** Pressure distribution nephogram of longitudinal section.

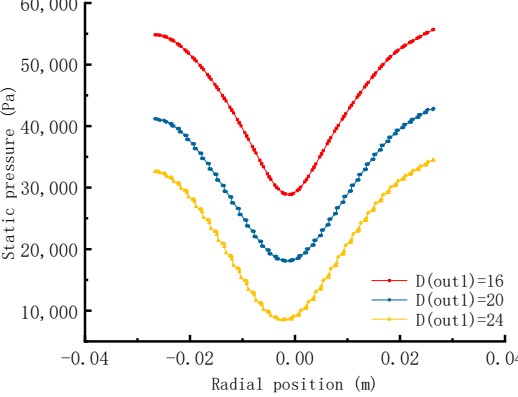

**Figure 4.** Static pressure distribution curve of section Z.

It can be seen from Figure 5 that the radial velocity distributions of the three cyclones were almost the same and distributed symmetrically; the tangential velocity presents an

"M2" shape, and the velocity gradually decreased as the position of the axis was approached. There was a maximum value on both sides of the center, and the tangential velocity was at a maximum when $D_{(out1)}$ = 24 mm and minimum when $D_{(out1)}$ = 16 mm, indicating that the ability of bubbles to flow toward the center in the radial position increased with an increase in the diameter of the exhaust port; the axial velocity was distributed axisymmetrically along the central axis. As it approaches the axis position, the velocity gradually increased. The central axial velocity was at a maximum when $D_{(out1)}$ = 16 mm. It was the smallest when $D_{(out1)}$ = 24 mm, which indicates that the ability of bubbles to flow toward the exhaust port in the axial position decreased with an increase in the exhaust port diameter. As the size of the exhaust port increased, if the gas outlet was in the positive direction, the axial velocity was convex at the center, which means that as the gas phase at the center flowed upward, liquid flowed downward on both sides.

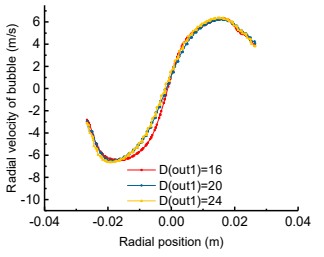 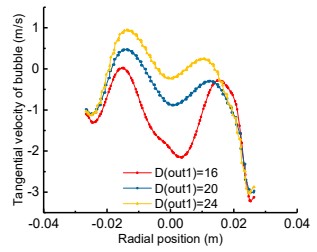 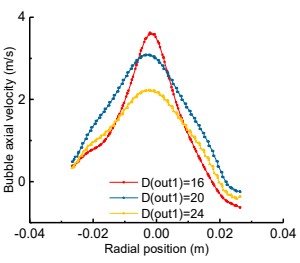

(**a**) Radial velocity distribution　　　(**b**) Tangential velocity distribution　　　(**c**) Axial velocity distribution

**Figure 5.** Section Z velocity distribution curve.

It can be seen from Figure 6 that the velocity distribution shows a spirally decreasing state, and the velocity gradient was relatively uniform. It can be inferred that when $D_{(out1)}$ = 24 mm, the trace was most concentrated at the bottom of the exhaust port, indicating the best bubble flow effect; when $D_{(out1)}$ = 16 mm, the velocity concentrated at the bottom of the exhaust port was low, indicating that the bubble flow effect was poor. From the velocity trace distribution of the bubble combined with the simulation analysis, it can be concluded that the gas discharge efficiency of $D_{(out1)}$ = 24 mm is 8% higher than that of $D_{(out1)}$ = 16 mm.

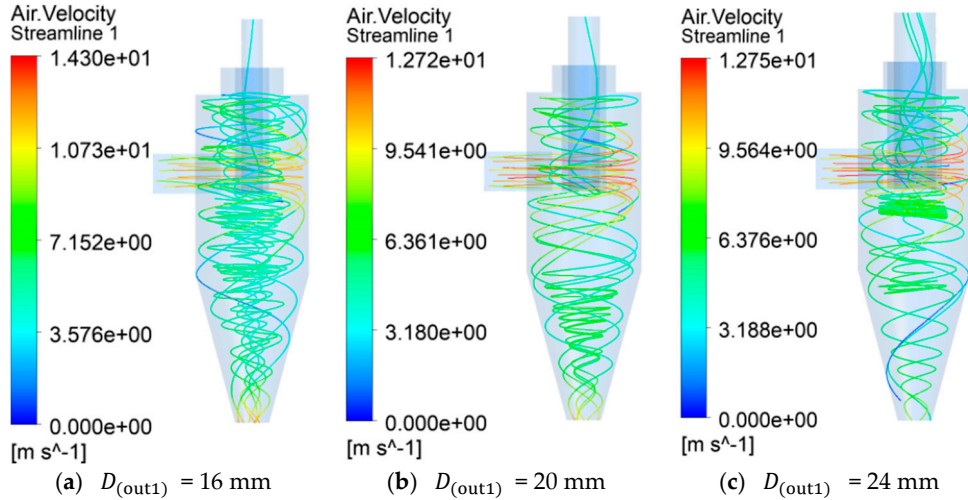

(**a**) $D_{(out1)}$ = 16 mm　　　(**b**) $D_{(out1)}$ = 20 mm　　　(**c**) $D_{(out1)}$ = 24 mm

**Figure 6.** Distribution of bubble motion in longitudinal section.

In order to explore the influence of the inlet's shape on the bubble's movement, the second scheme explored the three cases of rectangular and trapezoidal inlet cyclones and

trapezoidal inlet cyclones, as well as spiral inlet cyclones with spiral inlets, and its influence on the internal flow field.

It can be seen from Figure 7 that the cross-sectional pressure shows a layered decrease from the periphery to the middle area, and the overall pressure gradient shows an eccentric circle distribution. It can be concluded from Figure 8 that the pressure drop of the Z-section trapezoidal inlet was the largest; the drop between the rectangular and spiral inlets was not much different. The pressure drop of the trapezoidal inlet was five times that of the rectangular and spiral inlets, indicating that the trapezoidal inlet had a large pressure loss, which is not conducive to the flow of bubbles.

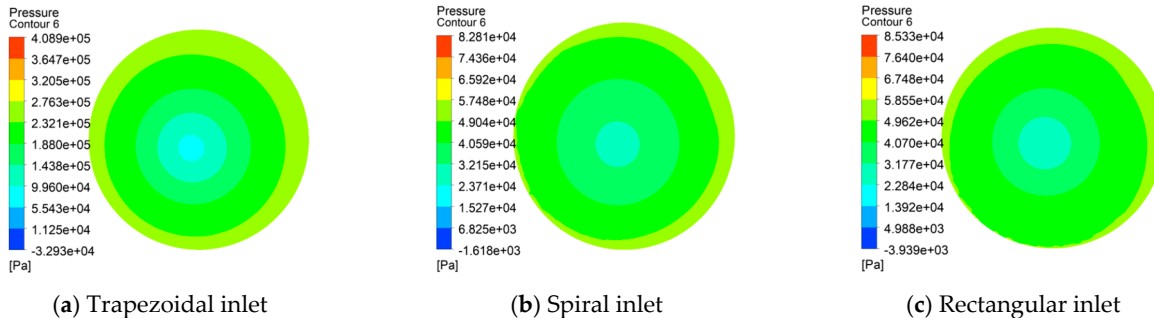

(**a**) Trapezoidal inlet        (**b**) Spiral inlet        (**c**) Rectangular inlet

**Figure 7.** Pressure distribution nephogram of section Z.

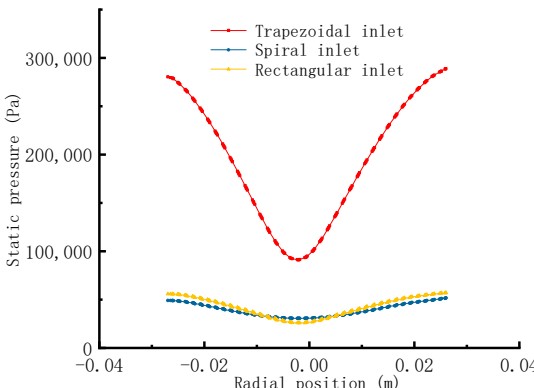

**Figure 8.** Static pressure distribution curve of section Z.

The tangential velocity is the cause of the centrifugal force, so the tangential velocity can be used to measure the separation effect. The tangential velocity has two peaks in the pressure distribution cloud chart, which are caused by the composite vortex formed by the internal and external vortices. This is the Ranking vortex [31].

Figure 9 shows the trend lines of the three sub-velocities in section Z. These three sets of parameters can well reflect the changing law for the flow field in the separator and reflect the vortex motion in the flow field. After the mixed liquid was separated by the swirling flow inside, the process of moving the bubbles upward and out was realized by the parameter of radial velocity. It is apparent from the radial velocity distribution curve in the figure that its distribution was basically symmetrical; the radial movement speed of the trapezoidal inlet bubble was the fastest, and that of the rectangular inlet bubble was the slowest. This shows that the trapezoidal inlet had the strongest ability to flow toward the center in the radial position, while the rectangular inlet had a relatively weak ability; theoretically, the greater the tangential velocity, the better the separation. From the tangential velocity curve in the figure, it can be observed that the tangential velocity was the largest at the trapezoidal inlet and the smallest at the spiral inlet. However, the overall tangential velocity profile of the trapezoidal inlet is extremely unstable and has no symmetry, indicating that the flow field at the trapezoidal inlet was extremely unstable,

which is not conducive to the swirling separation of bubbles. From the axial velocity distribution curve in the figure, it can be seen that the axial velocity distribution has a high middle and low sides. The velocity distribution of the spiral inlet is the lowest among the three structures as a whole, indicating that there was no strong swirling flow inside the spiral inlet cyclone.

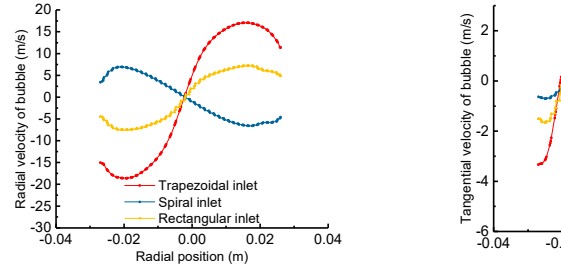
(**a**) Radial velocity distribution

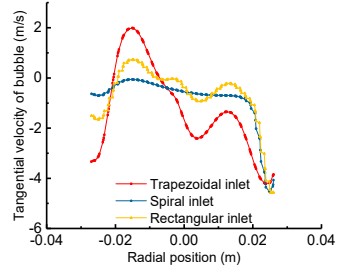
(**b**) Tangential velocity distribution

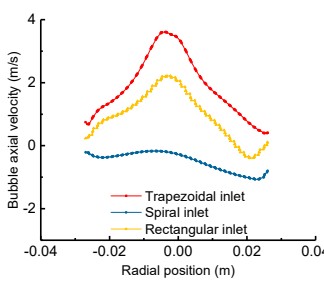
(**c**) Axial velocity distribution

**Figure 9.** Section Z velocity distribution curve.

It can be inferred from Figure 10 that when the inlet was rectangular, although the initial velocity was in the middle of the three, the trace was most concentrated at the bottom of the exhaust port, indicating that the bubble flow effect was the best. When the inlet was trapezoidal, although the maximum speed was the fastest, the concentration of the trace at the bottom of the exhaust port was slow, indicating that the bubble flow effect was poor. Combining the bubble velocity trace distribution and simulation analysis, it can be concluded that the gas discharge efficiency of the rectangular inlet was increased by 7% compared with that of the trapezoidal inlet cyclone.

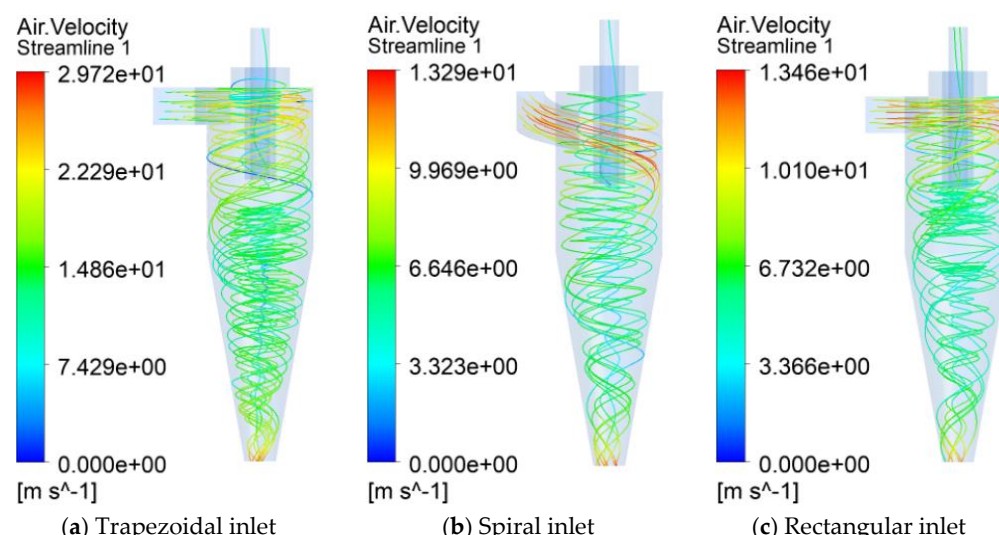
(**a**) Trapezoidal inlet      (**b**) Spiral inlet      (**c**) Rectangular inlet

**Figure 10.** Distribution of bubble motion in longitudinal section.

In order to explore the influence of the inlet position on the bubble's movement, in the third setup, a rectangular cyclone and rectangular central cyclone were selected for simulation analysis to explore their effects on the internal flow field.

At last, analyzing the static pressure distribution and velocity distribution curves of the rectangular inlet and the rectangular central inlet cyclone, it can be concluded that the inlet position had different effects on the bubble flow. The rectangular central inlet had a larger pressure drop than the rectangular inlet, and the bubble flow was faster; the overall flow effect was better.

As shown in Figure 11, a rectangular inlet cyclone with good simulation results was selected to simulate the flow of bubbles. In the initial state t = 0.00 s, the simulated oil entered the cyclone at a speed of 8 m/s, and no bubbles were generated at the beginning. When t = 0.07 s, a "candle flame" air column was formed from the center of the bottom of the discharge port to the center of the cyclone. At t = 0.12 s, the air column was connected to the exhaust port. The diameter of the air column near the exhaust port was smaller, and the diameter of the air column below the exhaust port was larger. When t = 0.20 s, the air column filled the exhaust port, forming a "sword"-shaped air column. Bubbles accumulated toward the center due to the centripetal force in the radial direction along with the oil swirl. At the same time, due to the influence of buoyancy, the bubbles flowed toward the exhaust port and formed a gas column. The flow simulation shows that the gas column moved from the discharge port to the exhaust port until the air column was filled with the exhaust port, and the overall bubble flow effect was obvious. Of the gas, 91% flowed to the exhaust port, indicating that the overall structure had a more obvious degassing effect.

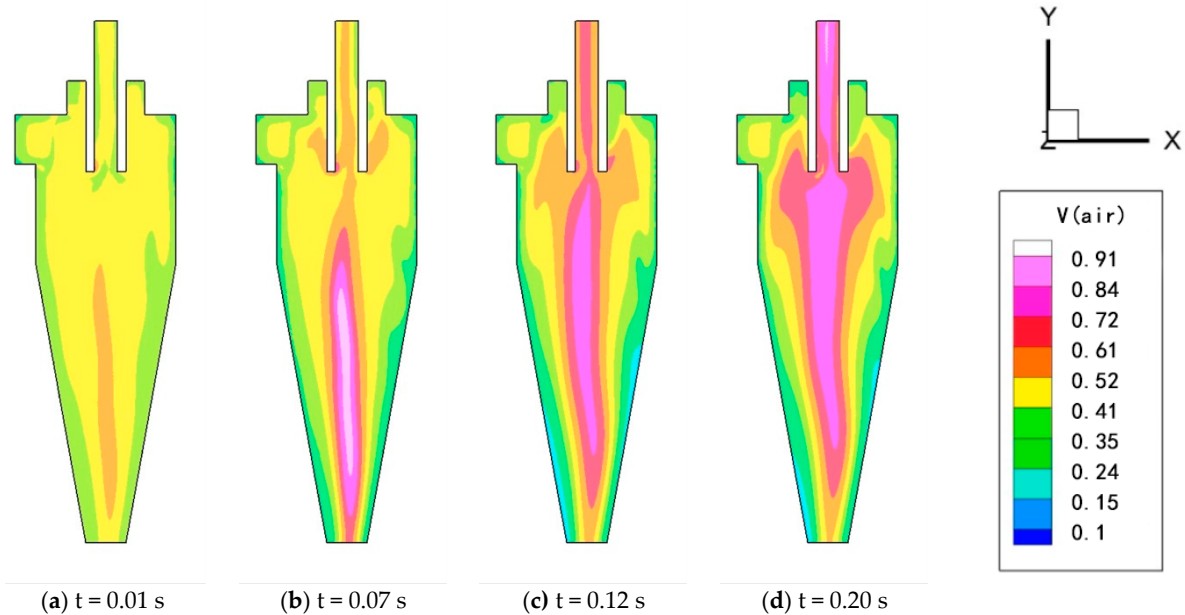

(**a**) t = 0.01 s     (**b**) t = 0.07 s     (**c**) t = 0.12 s     (**d**) t = 0.20 s

**Figure 11.** Simulation of bubble flow in cyclone.

## 3. Experimental Analysis

### 3.1. Design of Experimental Device

The rectangular inlet and spiral inlet cyclone with better bubble flow effects, according to the simulation analysis, were selected for experimental exploration. Figures 12 and 13 are the system schematic and physical diagrams of the constructed visual experimental platform. In order to visualize the bubble flow in the cyclone, the rectangular inlet and spiral inlet cyclones were designed and processed as shown in Figure 13a,b. The visible structure of the experimental system, such as the cyclone and the fuel tank, were all made from highly transparent acrylic material. During the experiment, the screw pump supplied oil to the system. The specific parameters are shown in Table 2. Experimental light was provided by a sheet light source. A high-speed camera with a macro lens was used to film different visualization areas to record the flow of bubbles.

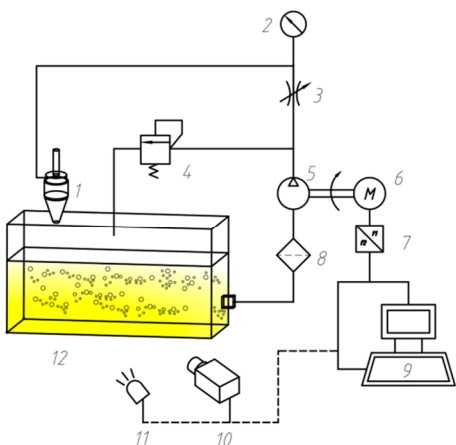

**Figure 12.** Experimental schematic diagram. 1—Gas–liquid cyclone separator; 2—Pressure gauge; 3—Throttle valve; 4—Safety valve; 5—Three screw pump; 6—Three-phase asynchronous motor; 7—Frequency converter; 8—Filter; 9—Computer; 10—High-speed camera; 11—Sheet light source; 12—Tank.

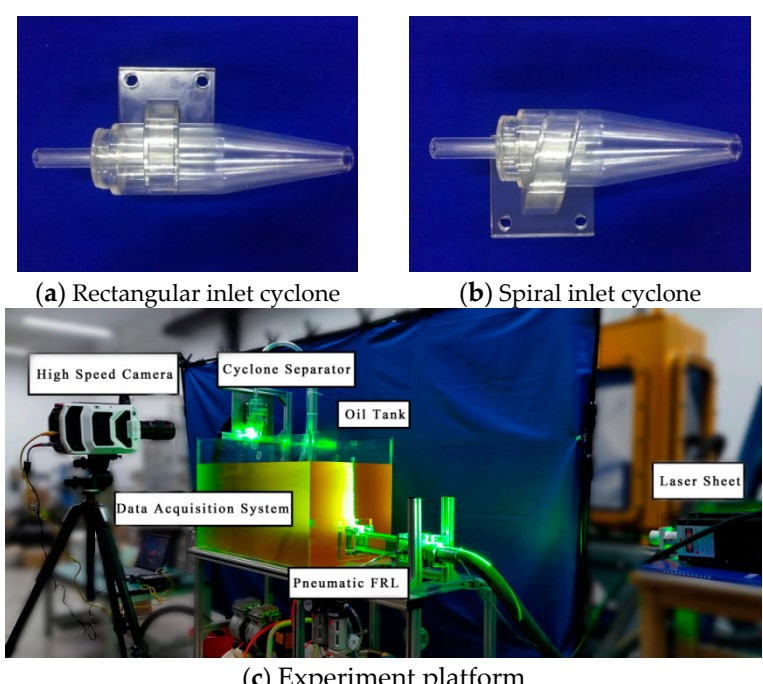

(**a**) Rectangular inlet cyclone      (**b**) Spiral inlet cyclone

(**c**) Experiment platform

**Figure 13.** Experimental platform for visualization of bubble flow in cyclone.

**Table 2.** Performance parameters of three-screw pump.

| Model | Flow (L/min) | Pressure (MPa) | Rated Speed (r/min) | Driving Power (KW) |
| --- | --- | --- | --- | --- |
| Hebei Far East Pump Co., L3GR30 * 4W2 | 60 | 1.0 | 2900 | 2.2 |

### 3.2. Analysis of Experimental Results

Figure 14 shows the visualized flow of bubbles in the rectangular inlet cyclone. Before the experiment, the oil in the system was in a static state. When the pump was started, the return oil entered the rectangular inlet cyclone at 8 m/s. Due to the influence of gravity, when the oil entered the inside of the cyclone, it formed asymmetrical liquid flow clusters along the inner wall. The oil swirled rapidly, and obvious bubbles of different sizes could

be observed on the inner wall. After 0.16 s, a visible funnel-shaped swirling air column gradually formed. Obvious bubbles gathered above the inside of the cyclone. The swirling air column gradually shrank until it had all overflowed from the exhaust port. After 0.7 s, the bubbles were arranged below the exhaust port of the cyclone to form an observable undulating line. With the continuous overflow of bubbles, large bubbles continued to accumulate above the inside of the cyclone.

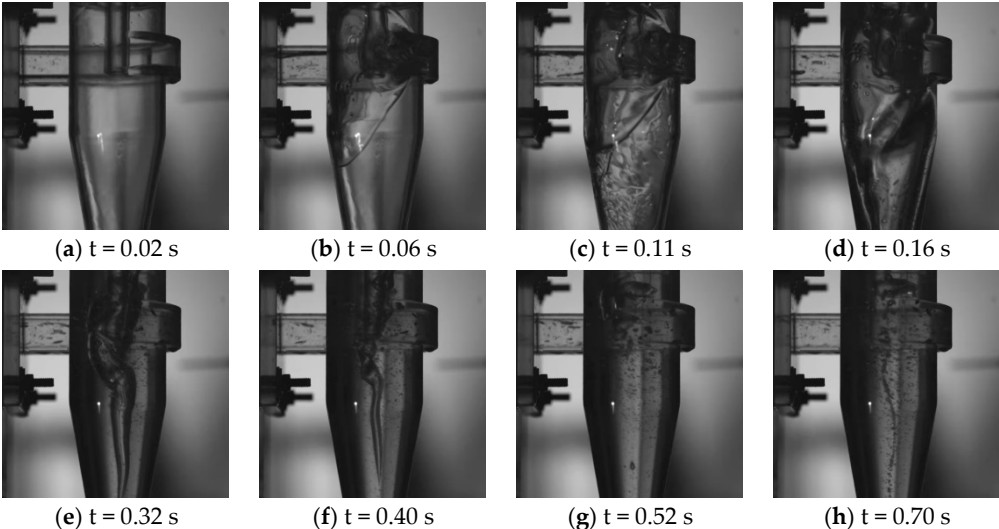

(**a**) t = 0.02 s    (**b**) t = 0.06 s    (**c**) t = 0.11 s    (**d**) t = 0.16 s

(**e**) t = 0.32 s    (**f**) t = 0.40 s    (**g**) t = 0.52 s    (**h**) t = 0.70 s

**Figure 14.** Bubble flow process in a cyclone with rectangular inlet.

Figure 15 shows the visualized flow of bubbles in the spiral inlet cyclone. The experimental conditions were the same as for the rectangular inlet cyclone. When the system return oil entered the spiral inlet cyclone, it was affected by gravity, forming an asymmetrical liquid stream along the inner wall with visible bubbles, and the oil rapidly swirled along the inner wall. The stratified funnel-shaped air column was obliquely erected, and the air column filled the upper part of the inside of the cyclone. The swirling air column gradually shrank until it had all overflowed from the exhaust port. After 0.89 s, the bubbles below the exhaust port of the cyclone were arranged to form a visible undulating line. As the bubbles continued to overflow throughout the entire swirling process, there was less accumulation of bubbles above the inside of the cyclone.

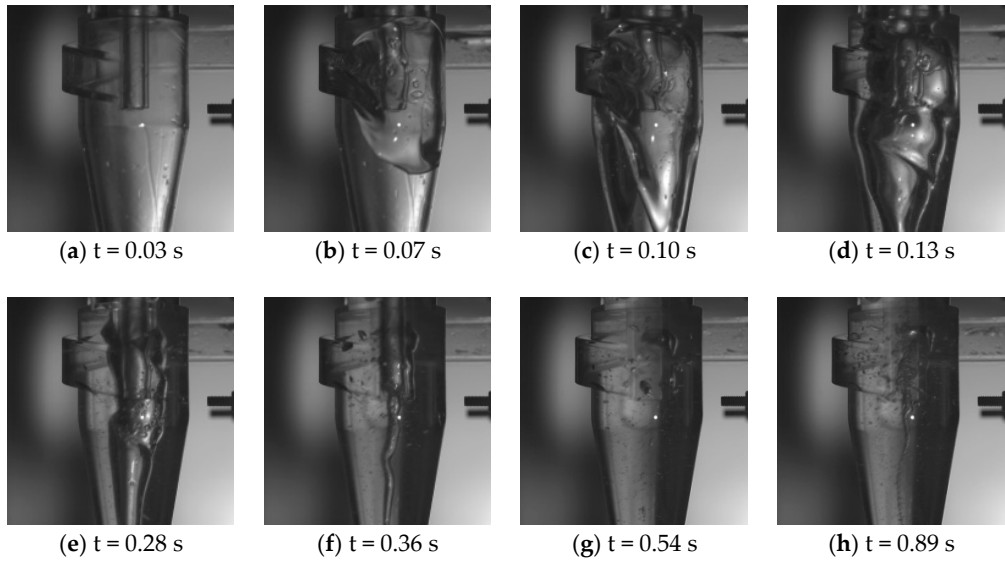

(**a**) t = 0.03 s    (**b**) t = 0.07 s    (**c**) t = 0.10 s    (**d**) t = 0.13 s

(**e**) t = 0.28 s    (**f**) t = 0.36 s    (**g**) t = 0.54 s    (**h**) t = 0.89 s

**Figure 15.** Bubble flow process in spiral inlet cyclone.

In the simulation results shown in Figure 11, only the bubbles gathered in the center of the cyclone to form a gas column and were discharged to the exhaust port. In the experimental process shown in Figure 14; however, the initial bubbles followed the swirling flow of the oil down the cyclone, and the gas column disappeared. There is also no process of forming an observable line. The reason for the difference between the analysis simulation and the experimental results may be due to the selection of the simulation model and the influence of the simulation parameters, which can be further studied in the later research.

When the bubble was in the rectangular and spiral inlets, it underwent a change from gradually forming a funnel-shaped air column to the air column disappearing and then forming a line of observable fluctuations. The rectangular inlet cyclone underwent oil-liquid separation more quickly than the spiral inlet cyclone; the overall bubble flow effect was better than that of the spiral inlet cyclone, and there were continuous large bubbles gathering above the inside. The spiral inlet cyclone did not exhibit the phenomenon of large bubbles gathering. The initial bubbles flowed downwards spirally along the outer wall of the cyclone due to the influence of gravity and the given initial velocity with the oil swirling flow. When the oil was filled with the liquid discharge port, the bubbles gathered towards the center due to radial centripetal force. The air bubbles flowed upward under the influence of buoyancy in the axial direction to form a gas column. Since the upper pressure was lower than the lower pressure, a funnel-shaped gas column formed. As the bubbles continued to flow to the exhaust port, the air column gradually collapsed until it disappeared. After a period of time, the remaining bubbles in the oil continued to swirl to the center, causing it to form a visible wave line near the center axis. The simulation result is only the process of forming an air column, and no observable fluctuating line was formed after the air column disappeared. The reason may be the fact that the transient excerpt was often short or the simulation model selected; more in-depth follow-up studies can be conducted.

## 4. Conclusions

The flow fields and bubble flow laws in cyclones with different structures were analyzed by comprehensive theoretical analysis, CFD numerical simulation and visual experimental research. The influence of the diameter of the cyclone's exhaust port, and the shape and position of the cyclone's inlet on the bubble flow was studied, and an experimental platform for the visual observation of bubble flow in the cyclone was built to observe and record the bubble flow process. The bubble flow mechanism was analyzed, and the main conclusions are as follows:

1. The exhaust port diameter had a great influence on the bubble flow effect. When the exhaust port diameter was 24 mm, the gas discharge efficiency increased by 8% compared with an exhaust port diameter of 16 mm. The larger the exhaust port diameter, the better the bubble flow effect.

2. The inlet shape of the cyclone had different effects on the bubble flow. When the rectangular inlet was selected, the initial bubble flow velocity was in the middle of the three, but the bubble flow effect was the best. When the trapezoidal inlet was selected, the maximum bubble flow speed was the fastest, but the bubble flow effect was poor. The gas discharge efficiency of the rectangular inlet was 7% higher than that of the trapezoidal inlet cyclone. The inlet position of the cyclone affected the bubble flow. The rectangular central inlet had a larger pressure drop than the rectangular inlet, the bubble flow was faster, and the overall flow effect was better.

3. When the bubble was in the rectangular and spiral inlets, it underwent a change from gradually forming a funnel-shaped air column to the air column disappearing and then forming an observable fluctuating line. The rectangular inlet cyclone underwent gas–liquid separation for a shorter period of time than the spiral inlet cyclone; the overall bubble flow effect was better than that of the spiral inlet cyclone, and large bubbles continued to accumulate above the inside of the cyclone. The spiral inlet cyclone did not exhibit the phenomenon of large bubbles gathering.

　　　　The experimental results verify the feasibility of the relevant structural design and the rationality of the conclusions drawn from the numerical simulation, and more deeply elucidate the mechanism of bubble flow.

**Author Contributions:** Conceptualization, supervision, project administration, funding acquisition, H.J.; methodology, data curation, writing—original draft preparation, Y.B.; software, validation, visualization, Y.L.; investigation, formal analysis, writing—review and editing, L.L.; software, formal analysis, S.Y. All authors have read and agreed to the published version of the manuscript.

**Funding:** This research was funded by the National Natural Science Foundation of China under grant number 51575254 and in part by the Special Fund Project of Strategic and New Industry Development of Jiangsu Province—The Research and Industrialization of Core Technology of Hydraulic Pump and Valve Applied in High-End Construction Machinery.

**Data Availability Statement:** Data is contained within the article.

**Conflicts of Interest:** The authors declare no conflict of interest.

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
