# Peer review of "Analysis of Bubble Flow Mechanism and Characteristics in Gas–Liquid Cyclone Separator"

_processes, doi:10.3390/pr9010123_

Round 1

Reviewer 1 Report

The paper concerns the liquid-gas separators based on the cyclone-type design. Numerical and experimental studies are reported, with varying cyclone separator geometry and flow conditions.

The paper contains some interesting results, although these is significant room for improvement. The following particular issues need to be resolved in the revision:

  1. Page 2: the phrase "Mikheev et al. conducted numerical and experimental studies on the structural parameters of the cyclone and its degassing efficiency [12-15]." must be rewritten because only [12] belongs to Mikheev.
  2. Section 2 of the paper contains some results obtained on the basis of the second Newton's law for a bubble. Such an analysis is a kind of common knowledge now, the conclusion drawn from it are trivial (larger bubbles are separated better than the fine ones). Also, in Fig.2 the curves are colored wrongly (the upper one must correspond to the largest diameter). The last paragraph on pages 4-5 is very difficult to read, it contains some speculations not supported by the previous contents. In my view, all this section can be omitted because it does not give any meaningful information for the subsequent paper content.
  3. The numerical and experimental parts are more interesting and meaningful, these results may well be of interest to specialists in the design of cyclone gas-liquid separators.
  4. English language can be improved in many places

Author Response

Dear reviewer, I have carefully read and revised your valuable comments.

Reviewer 2 Report

The submitted manuscript presented a simulation analysis followed by an experimental analysis through flow visualization. The research is mostly based on the prediction of flow behavior for different designs of cyclone separators. The investigation has reasonable scientific merit for publication. However, the following points are suggested for improving the manuscript prior to publication.

  • The different geometry of cyclones and boundary conditions were presented. However, meshing information and grid convergence information are not presented in the manuscript to support numerical modeling.
  • Although, the paper presented numerical modeling and experimental flow visualization. However, the authors are requested to add a validation diagram for their numerical model to enhance the quality of the paper.
  • The governing equations solved for the numerical simulations would provide a better understanding of the numerical simulations performed.
  • Some figures are not mentioned in the discussion, for example, Figure 3 and Figure 4. Further, the authors are requested to maintain the uniformity of the presentation of figure numbering. Somewhere it is mentioned as Figure 9 and again it is mentioned as Fig. 10.
  • A section with future scopes of this research would enlighten the readers on the potential research directions.
  • Careful Proofreading would reduce the minor grammatical mistakes or typos in the paper.

Author Response

(The authors gave the same response as above.)

Round 2

Reviewer 1 Report

I appreciate the authors' efforts which improved the paper in many places. However, there are still significant issues which must be resolved before the paper can be published.

  1. In my first review I recommended to eliminate Section 2 completely as not giving any significantly new information. The authors decided to keep it in the paper, evidently considering the analysis of single bubble motion an important part of the paper. Ok, but then we need to take a deeper look into this analysis. The main objection to the approach, in my view, is that the horizontal and vertical components of bubble velocity are considered independently: uc is obtained from Eqs. (1)-(3) (recast as Eq. (4)), while ua is obtained from Eqs. (5)-(9), recast as Eq. (10). However, Eqs. (4) and (10) are not independent and cannot be solved as such: the coupling is achieved via the drag force which must be written in the form of Eq.(7):

F=-1/2CDρ0πRa2|ub-u|(ub-u)

where boldface character denote vector quantities, ub=(uc,ua). Even if the velocity of liquid is zero, u=0, the absolute value |ub|=(uc2+ua2)1/2, i.e., this  factor cannot be taken equal to uc in equation (3) and equal to ua in Eq.(7), in both these equations this term must be taken the same. Also, CD depends on the Reynolds number based on the relative velocity that also couples the radial and axial velocity components. In the case of Stokes law, the components may decouple, but the derivation of this must be given explicitely, not by writing wrong general equations.

2. It is not clear why the vertical momentum equation contains more forces than the horizontal one (Eqs (7) and (8))?

3. Nowhere in Section 3 any information on the "oil" velocity u is given. What was its value in the calculations presented in Figure 3? Was it zero? Then why write it at all? Was it non-zero? Then at which value the data are obtained? And what happens at larger or smaller values of u?

4. The same concerns Figure 2, at which angular velocity the results are obtained? What happens if the angular velocity is higher or lower? Presenting three curves for unknown parameters is meaningless, and no conclusions can be derived therefore.

5. The authors improved English, however, still there are remaining inconsistencies:

"movement rate" - must be "velocity component", or speed - check the whole Section 2 for these phrases

Section 3: "The number of grids in this study was 1,449,887" - There was only one grid in the simulation, and grid contained 1,449,887 cells. A grid (or mesh) is a collection of cells in CFD. Check the whole text.

Author Response

(The authors gave the same response as above.)
